# Effects of Starvation on the Physiology and Liver Transcriptome of Yellowcheek (*Elopichthys bambusa*)

**Min Xie** [1], **Shaoming Li** [1], **Zhifeng Feng** [1], **Jin Xiang** [1,2], **Qi Deng** [1], **Pengpeng Wang** [1], **Hao Wu** [1], **Jingwei Gao** [1], **Guoqing Zeng** [1,*] **and Guangqing Xiang** [3]

1 Hunan Fisheries Science Institute, Changsha 410153, China
2 Hunan Aquatic Foundation Seed Farm, Changsha 410153, China
3 Longshan County Animal Husbandry and Aquatic Affairs Center, Xiangxi 416800, China
* Correspondence: zengguoqing001@163.com

**Abstract:** Anthropogenic and extreme climate disasters cause ecological changes in natural rivers and lakes, increasing the risk of starvation in yellowcheek (*Elopichthys bambusa*). Therefore, the impact of starvation on the metabolism and wild population of yellowcheek should be explored. In this study, we used transcriptome sequencing technology to analyze the effects of short (8 d) and long-term (28 d) starvation on the liver transcriptome, growth, and serum indicators of yellowcheek. Our results showed that short-term starvation significantly reduced the visceral weight and viscera index of yellowcheek. Long-term starvation significantly reduced the body weight and Fulton's condition factor, and it maintained significant reductions in visceral weight and viscera index. These results indicate that glycogen is the preferred energy source, rather than muscle protein, under starvation. Short-term starvation limited N-glycan and fatty acid biosynthesis, fatty acid elongation in the endoplasmic reticulum in the liver, and upregulated fatty acid degradation. However, long-term starvation alleviated the reduction in N-glycan and fatty acid biosynthesis caused by early starvation, and it significantly reduced fatty acid elongation in the mitochondria, as well as fatty acid degradation. These results provide important experiment information for assessing the starvation levels and nutritional status of wild yellowcheek.

**Keywords:** starvation; *Elopichthys bambusa*; liver transcriptome; growth

**Key Contribution:** Effects of short- (8 d) and long-term (28 d) starvation on the liver transcriptome; growth; and serum indicators of yellowcheek were analyzed and these information can be used to assess the starvation levels and nutritional status of wild yellowcheek.

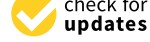



## 1. Introduction

Starvation is tolerated by many fish species throughout their life history [1,2]. A restricted diet causes stress to fish, which may further affect growth and other physiological functions [3]. For instance, Machado et al. [4] reported that blood sugar levels of catfish (*Rhamdia hilarii*) declined, progressively, to about 50% of fed values after starvation of 30 d, whereas plasma-free fatty acid concentration increased twofold. Moreover, they also found that starvation reduced concentrations of lipid and glycogen in the liver and of glycogen, lipid, and protein in white muscle. Raina and Sachar [5] found that starvation of 60 d reduced macrophages, lymphocytes, monocytes, neutrophils, eosinophils, and basophils in the spleen and kidney of *Labeo boga*, whereas thrombocytes depicted an increase in their count. To survive during periods of unfavorable feeding conditions, fish reduce their energy expenditure derived from protein turnover [6] and mobilize their endogenous reserves to obtain the energy required to maintain vital processes [1].

Although the metabolic characteristics of fish under starvation have been extensively studied, owing to the heterogeneity of fish (such as feeding habits, fish age, and nutritional

stage) and their habitat diversity, a unified theoretical framework on the effects of starvation on fish metabolism is not yet established [1]. For instance, liver glycogen is the first energy reserve mobilized to face starvation, and lipid reserves are used to obtain energy in parallel with glycogen mobilization in some fishes. Then, muscle protein is the last reserve mobilized [7,8]. However, liver glycogen are preserved in some fishes, proteins are degraded for gluconeogenesis, and lipids and/or proteins are first used as energy substrates [7,9]. For instance, Machado et al. [4] estimated that most of the energy utilized by catfish during starvation derived from the catabolism of muscle lipid and protein. Sturgeon (*Acipenser naccarii*) displayed an early increase in hepatic glycolysis during starvation, whereas trout (*Oncorhynchus mykiss*) preferentially used lactate for gluconeogenesis in liver and white muscle [1]. Moreover, whether ketone bodies are an important energy source during fish starvation remains under debate [1]. For instance, sturgeon exhibited a higher lipid-degradation capacity and greater synthesis of hepatic ketone bodies than trout, although the latter species also exhibited strong synthesis of ketone bodies during starvation [1].

Yellowcheek (*Elopichthys bambusa*) is a large carnivorous fish confined to Asia [10–12]. As one of the species at the top of the aquatic ecosystem food chain, yellowcheek plays an important role in controlling fish community structure and maintaining balance in the aquatic ecosystem [13,14]. Recently, because of the isolation of rivers and lakes, construction of irrigation, and environmental depletion of aquatic ecosystems, the natural population of yellowcheek has declined rapidly in China; thus, this species is scarcely found in large populations in rivers and lakes [12,14], and it is listed on the International Union for Conservation of Nature (IUCN) Red List of threatened species (Version 2014.1) [15]. Therefore, this species should be protected through conservation strategies and management. To establish effective conservation and sustainable management strategies for wild yellowcheek populations, the ecological, genetic, and physiological characteristics of this organism should be systematically studied. Although studies on wild yellowcheek resources [13,14] and genetic diversity [12,16] are available, the effect of hunger on the metabolism of yellowcheek has not been systematically studied. Considering that changes in natural river and lake ecosystems, caused by human activities and extreme climate and geological disasters, may lead to an increase in the risk of starvation in yellowcheek, the impact of starvation on the metabolism of wild yellowcheek should be explored. Moreover, although the anatomical method (filling degree) is usually used to evaluate whether fish diet is rich [17], this method cannot evaluate the time of fish starvation in the case of diet shortage. The analysis of the differences in the serum parameters and liver transcription of fish with different starvation levels will provide an important way to assess the starvation level of natural fish.

Therefore, to clarify the effect of starvation on the metabolism of yellowcheek, as well as to find the serial parameters and liver transcriptional markers used to evaluate the different starvation levels of yellowcheek, in this study, we used transcriptome sequencing technology to analyze the effects of starvation on the liver transcriptome, growth, and serum indicators of yellowcheek for 8 and 28 d. Our results provide important reference data for the follow-up assessment of the impact of starvation on wild yellowcheek.

## 2. Materials and Methods

### 2.1. Fish and Experimental Design

Yellowcheek were provided by the Hunan Fisheries Research Institute. A total of 360 healthy and nearly sized fish, with an average body weight of 221.36 ± 6.75 g and average body length of 33.09 ± 1.33 cm, were selected from an outdoor pond. They were randomly and equally assigned to 12 outdoor cement pools (10 m × 5 m × 1 m) with 0.8 m of water depth, to acclimatize for 1 week before the formal experiment, and they were fed commercial feed (crude protein ≥48%, crude fat ≥5.0%, lysine ≥2.8%, moisture ≤10%, and ash ≤18%) twice daily (8:00 and 18:00). The dry weight of the daily feed was 3% of the average body weight of the fish as measured before the experiment.

In each group, two control groups, two treatment groups, and three replicates were established. The treatment groups were starved for 8 (F8) and 28 d (F28), while the control groups were continuously fed a commercial diet (F8C and F28C) in the acclimatization stage. During the experiment, the water temperature, pH, and dissolved oxygen were $23.5 \pm 3.4$ (from 20.2 to 27.6) $^\circ$C, $6.9 \pm 0.2$ (from 6.7 to 7.2), and $6.1 \pm 0.3$ (from 5.7 to 6.5) mg/L, respectively. A small aerator was used in each cement pool to continuously increase oxygen during the experiment.

### 2.2. Sample Collection and Preservation

There were five fish from each group sampled. Fish were anesthetized with MS222, and then, the body weight and length of each fish were measured. Whole blood was collected from the caudal vein, the serum was left to stand at 4 $^\circ$C for 12 h, and the resulting supernatant was collected. After dissection, the visceral weight of each fish was measured, and then, the liver was collected, quickly frozen in liquid nitrogen, and stored at $-80$ $^\circ$C.

### 2.3. Determination of Serum Indicators

The levels of alanine aminotransferase (ALT), aspartate aminotransferase (AST), lactate dehydrogenase (LDH), glucose (GLU), serum total protein (STP), albumin (ALB), alkaline phosphatase (ALP), triglyceride (TG), total cholesterol (TC), high-density lipoprotein (HDL), and low-density lipoprotein (LDL) were determined using a Chemray 800 automatic biochemical analyzer (Leidu Life Science and Technology, Shenzhen, China).

### 2.4. RNA Extraction and Transcriptome Sequencing

Total RNA was extracted from 0.2 g liver tissue from each fish using the mirVana miRNA isolation kit (Ambion, Austin, TX, USA), following the manufacturer's protocol. RNA integrity was evaluated using the Agilent 2100 Bioanalyzer (Agilent Technologies, Santa Clara, CA, USA). Sequencing libraries were then constructed using the TruSeq Stranded mRNA LTSample Prep kit (Illumina, San Diego, CA, USA) according to the manufacturer's instructions. Subsequently, the libraries were sequenced on the HiSeq X Ten platform (Illumina, San Diego, CA, USA), and 150 bp paired-end reads were generated. Transcriptome sequencing and bioinformatics analyses were conducted by OE Biotech Co., Ltd. (Shanghai, China). Raw reads were processed using Trimmomatic [18]. Reads containing poly-N and low-quality reads were removed to obtain clean reads. After removing the adaptor sequence, the clean reads were assembled into expressed sequence tag clusters and de novo assembled into transcripts using Trinity 2.4 [19] and the paired-end method. The longest transcript was chosen as a unigene, based on sequence similarity and length, for subsequent analysis. The unigene sequences were aligned with the NCBI nonredundant (NR), SwissProt, and clusters of orthologous groups for eukaryotic complete genome (KOG) databases using BlastX (threshold E-value: $10^{-5}$) [20] for functional annotation. Gene ontology (GO) classification was used to map the relationship between SwissProt and GO terms. The unigenes were also mapped to the Kyoto Encyclopedia of Genes and Genomes (KEGG) database [21] to annotate potential metabolic pathways. The read counts and fragments per kilobase of exon model per million mapped fragments (FPKM) [22] values of each unigene were calculated using bowtie2 [23] and eXpress [24]. Differentially expressed genes were identified using the R-DESeq package [25]. A value <0.05 and foldChange >2 or foldChange <0.5 were set as the threshold for significantly differential expression. GO enrichment and KEGG pathway enrichment analyses of differentially expressed genes were performed using R based on hypergeometric distribution.

### 2.5. Data Analysis

Fulton's condition factor (K) was calculated, as described previously [26]. The viscera index was calculated as the ratio of the viscera to body weight. Principal component analysis (PCA) was conducted using the R vegan package. The Kruskal–Wallis rank sum

test with the Dunn post-hoc test was used for statistical testing of the data. $p < 0.05$ was considered as significant.

## 3. Results

### 3.1. Effects of Starvation on Growth and Serum Indices of Yellowcheek

No fish died during the experiment. Starvation for 8 d did not cause a significant difference in the body weight of yellowcheek compared with that in the control ($p > 0.05$; Figure 1A). Meanwhile, starvation for 28 d significantly limited yellowcheek growth, resulting in a significantly lower body weight than that of the control ($p < 0.01$; Figure 1A). Although starvation also limited the increase in body length, no significant difference was observed compared with that of the control ($p > 0.05$; Figure 1B). Starvation for 8 d reduced the K of yellowcheek, albeit no significant difference was observed compared with that of the control ($p > 0.05$; Figure 1C). However, K after 28 d of starvation was significantly lower than in the control ($p < 0.01$; Figure 1C). In addition, compared with the control, starvation for 8 d significantly reduced the visceral weight, viscera index, and liver weight, and starvation for 28 d maintained a significant reduction in these parameters ($p < 0.05$; Figure 1D–F). These results showed that starvation significantly reduced the body weight, K, visual weight, viscera index, and liver weight of yellowcheek. The decrease in visual weight, viscera index, and liver weight showed a significant difference earlier than the decrease in body weight and K, indicating that yellowcheek preferentially used glycogen to provide energy, rather than muscle protein, under starvation.

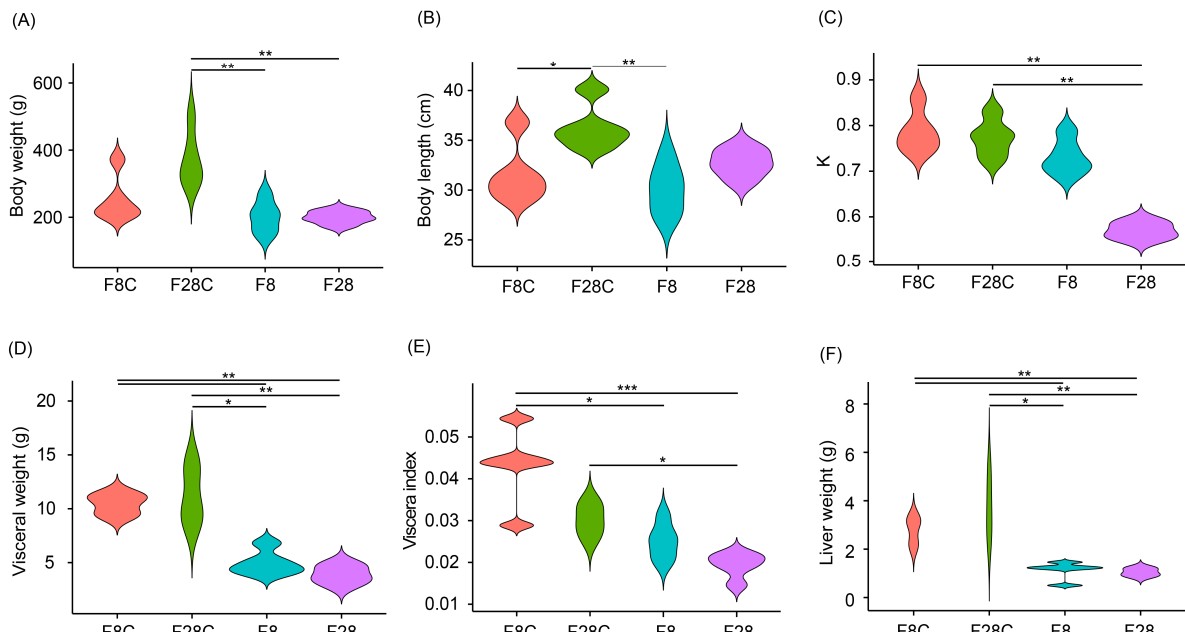

**Figure 1.** Effects of starvation on yellowcheek morphology. (**A**) Body weight, (**B**) body length, (**C**) Fulton's condition factor (K), and (**D**) visceral weight; (**E**) viscera index calculated as ratio of viscera and body weight; (**F**) liver weight. F8C: un-starved yellowcheek samples collected on the 8th day after the starvation experiment; F28C: un-starved yellowcheek samples collected on the 28th day after the starvation experiment; F8: starved yellowcheek samples collected on the 8th day after starvation experiment; F28, starved yellowcheek samples collected on the 28th day after the starvation experiment. * $p < 0.05$; ** $p < 0.01$; *** $p < 0.001$.

Compared with the control, starvation for 8 d did not significantly affect the serum GLU, STP, globulin, ALB, and HDL levels in yellowcheek, whereas the concentrations of these serum indicators, after starvation for 28 d, were significantly lower than those in the control ($p < 0.05$; Figure 2A,D–F,I). The serum ALT, AST, TG, TC, and ALP levels were significantly reduced after 8 d of starvation, and this significant reduction was maintained

after 28 d of starvation ($p < 0.05$; Figure 2B,C,G,H,K). The serum LDL and LDH concentrations were not significantly affected by starvation for 8 and 28 d ($p > 0.05$; Figure 2J,L).

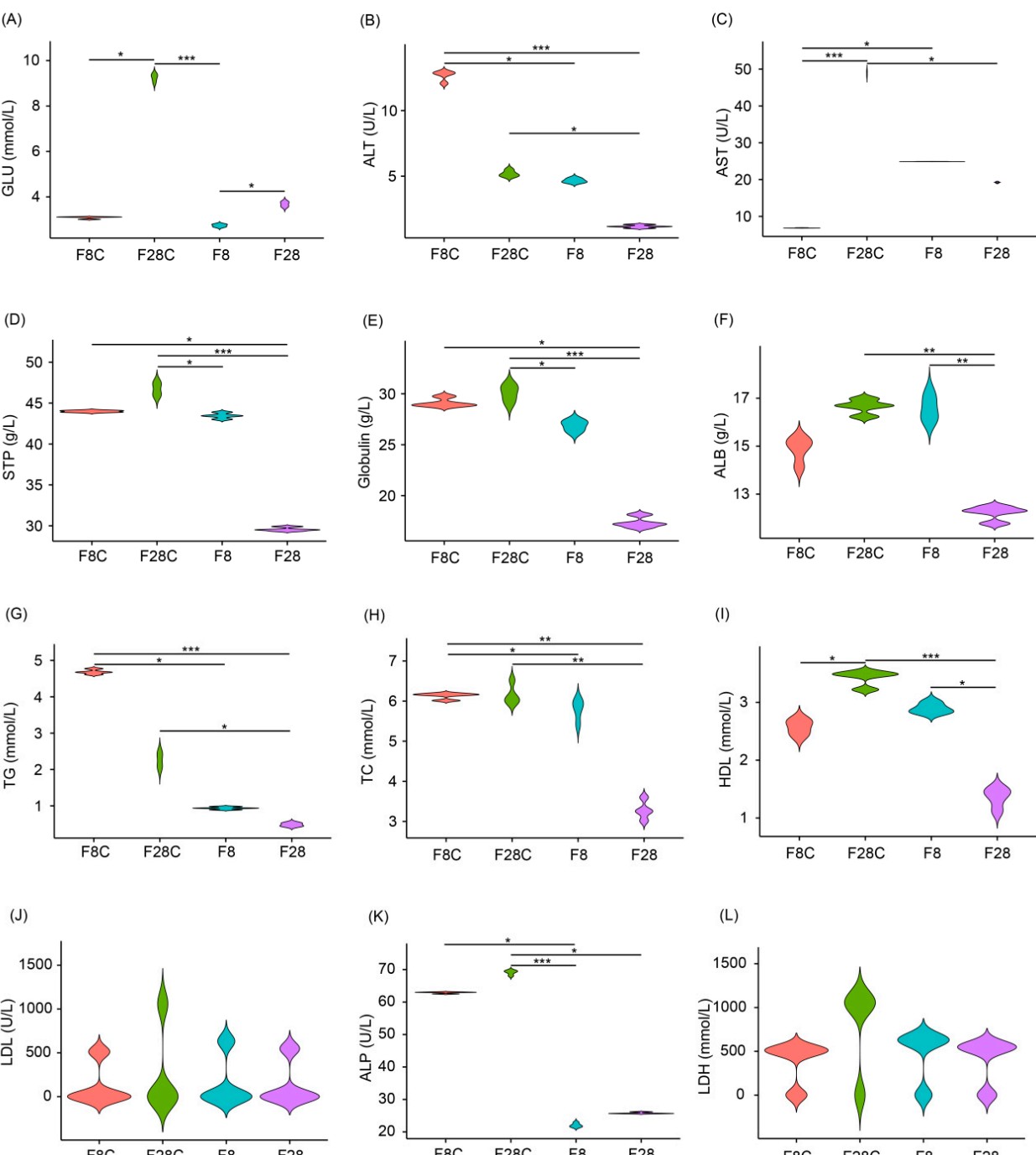

**Figure 2.** Effects of starvation on serum indices of yellowcheek. (**A**) Serum glucose (GLU), (**B**) alanine aminotransferase (ALT), (**C**) aspartate aminotransferase (AST), (**D**) serum total protein (STP), (**E**) globulin, (**F**) albumin (ALB), (**G**) triglyceride (TG), (**H**) total cholesterol (TC), (**I**) high-density lipoproteins (HDL), (**J**) low-density lipoproteins (LDL), (**K**) alkaline phosphatase (ALP), and (**L**) lactate dehydrogenase (LDH). F8C: un-starved yellowcheek samples collected on the 8th day after the starvation experiment; F28C: un-starved yellowcheek samples collected on the 28th day after the starvation experiment; F8: starved yellowcheek samples collected on the 8th day after the starvation experiment; F28: starved yellowcheek samples collected on the 28th day after the starvation experiment. * $p < 0.05$; ** $p < 0.01$; *** $p < 0.001$.

### 3.2. Effect of Starvation on Liver Transcriptome of Yellowcheek

To analyze the effect of starvation on the liver transcriptome of yellowcheek, transcriptome sequencing of 12 samples (three replicates in each group) were performed, and 81.68 GB of clean data were analyzed. The effective data volume of each sample was 6.28–7.15 GB, the $Q_{30}$ was 94.15–95.11%, and the average GC content was 47.14% (Table S1). A total of 42357 unigenes were spliced, with a total length of 40435640 bp and an average length of 954.64 bp. Furthermore, 27,574 (65.10%), 22,702 (53.60%), 10,562 (24.94%), 17,432 (41.15%), 25,056 (59.15%), 20,385 (48.13%), and 17,103 (40.38%) unigenes were annotated using the NR, Swissprot, KEGG, KOG, eggnog, gene ontology (GO), and Pfam databases, respectively. The ratio of reads compared to unigenes was 81.55–88.24%. A total of 29,887 coding sequences (CDS) were predicted, including 27,588 and 2299 sequences predicted using the database comparison and method, as well as ESTScan, respectively. Compared to the corresponding control samples, the number of differential genes detected in the samples starved for 8 and 28 d were 7647 and 3390, respectively (Figure 3A and Figure S1). PCA also showed that starvation significantly altered the liver transcriptome of yellowcheek (Figure 3B).

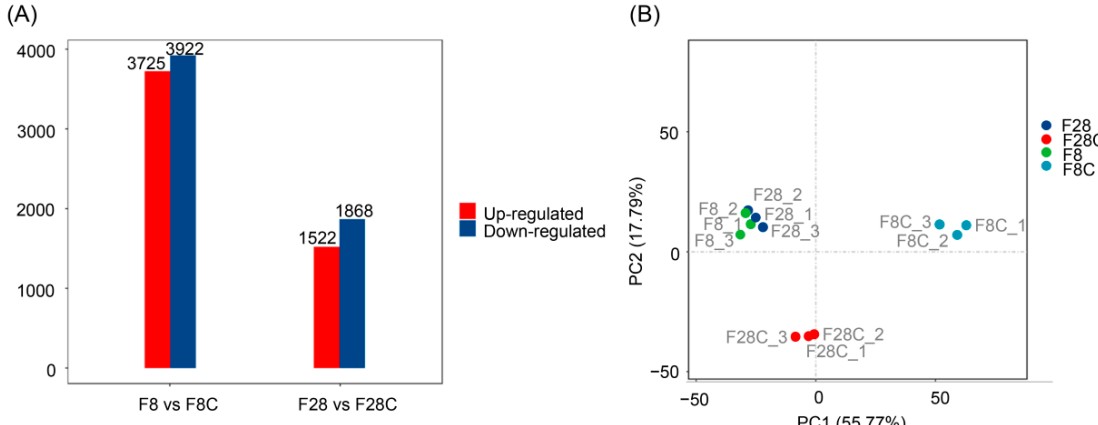

**Figure 3.** Effects of starvation on the liver transcriptome of yellowcheek. (**A**) The number of significantly upregulated and downregulated genes after 8 and 28 d of starvation. (**B**) Principal component analysis profile showing the differences in the liver transcriptome of yellowcheek with or without starvation. F8C: un-starved yellowcheek samples collected on the 8th day after the starvation experiment; F28C: un-starved yellowcheek samples collected on the 28th day after the starvation experiment; F8: starved yellowcheek samples collected on the 8th day after the starvation experiment; F28, starved yellowcheek samples collected on the 28th day after the starvation experiment.

GO annotation results showed that, after starvation, significant differential genes were mainly involved in biological adhesion, biological regulation, cellular component organization or biogenesis, cellular processes, developmental processes, establishment of localization, growth, immune system processes, localization, locomotion, metabolic processes, multi-organism processes, multicellular organismal processes, negative/positive regulation of biological processes, regulation of biological processes, reproduction, reproductive processes, responses to stimuli, rhythmic processes, signaling, cell junctions, extracellular matrix, extracellular region, macromolecular complex, membrane, membrane-enclosed lumen, organelle, synapse, binding, catalytic activity, enzyme regulator activity, molecular transducer activity, receptor regulator activity, structural molecule activity, translation regulator activity, and transporter activity (Figure 4A). Moreover, changes in the liver transcriptome of yellowcheek starved for 8 and 28 d were similar (Figure 4).

(A)

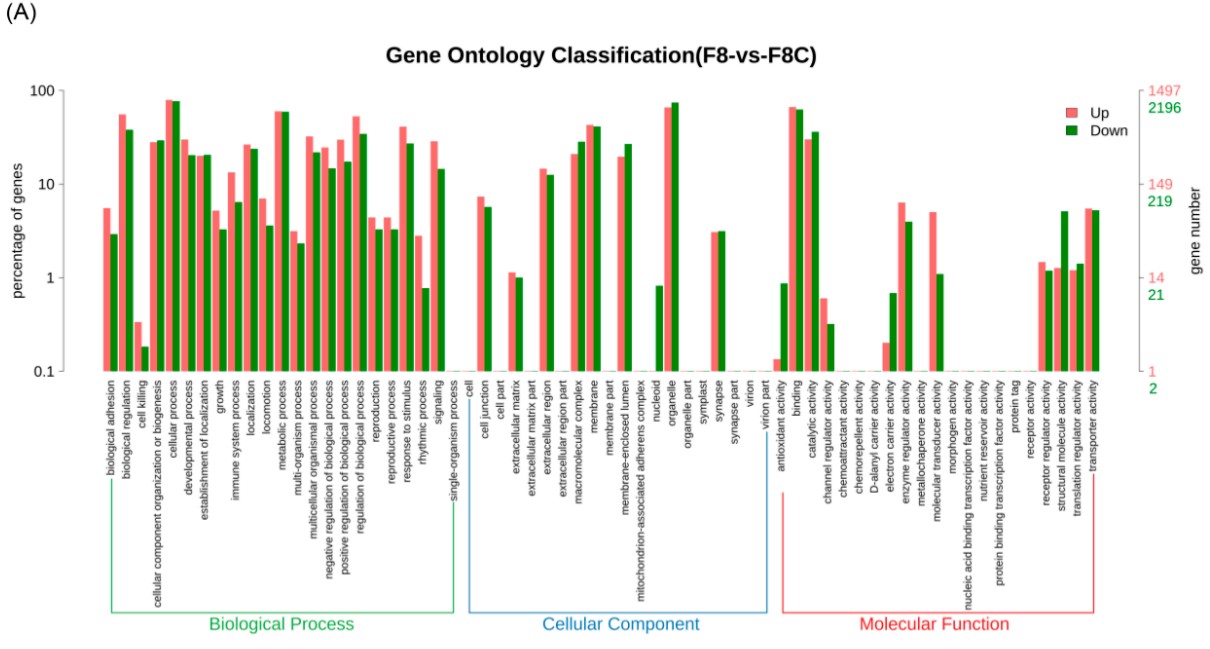

(B)

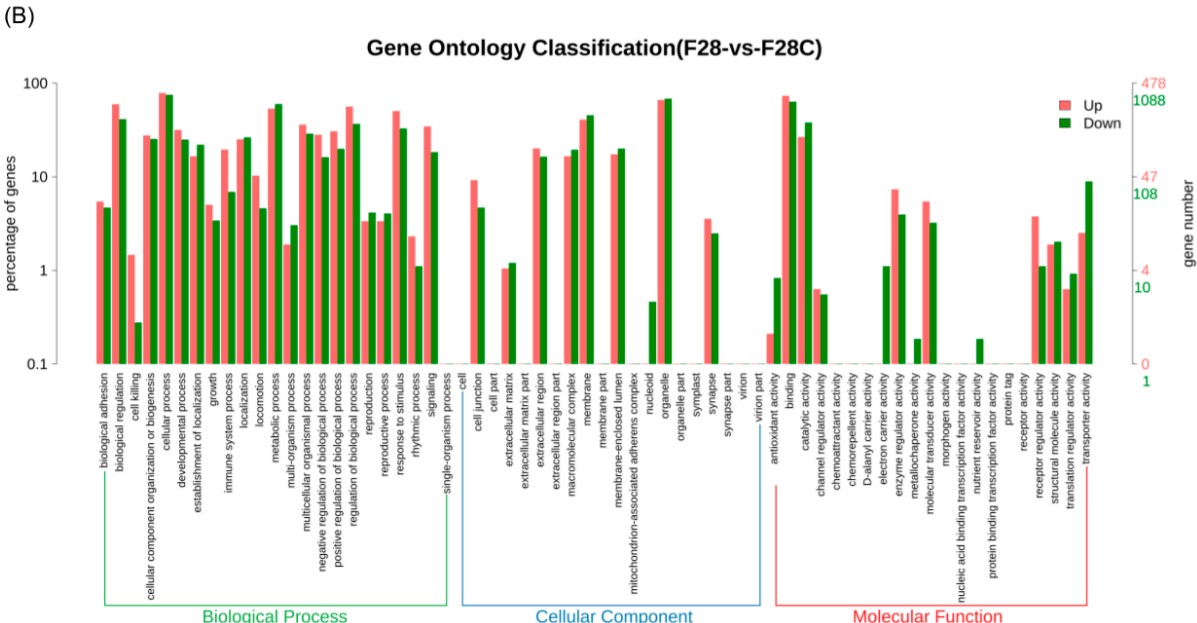

**Figure 4.** Gene ontology classification of significantly upregulated and downregulated genes in the liver transcriptome of yellowcheek starved for (**A**) 8 and (**B**) 28 d compared with the control groups at the same time.

KEGG annotation results showed that genes with significant changes at the transcriptional level, caused by starvation, mainly participated in cellular processes, environmental information processing, genetic information processing, metabolism, and organismal systems. Interestingly, genes involved in human diseases also exhibited significant changes at the transcriptional level. The pathways involved in these genes may involve human diseases and other metabolic and immune processes simultaneously. Moreover, the results observed in yellowcheek starved for 8 and 28 d were similar (Figure 5). It is also worth noting that the number of genes involved in metabolism and genetic information processing that significantly decreased during starvation was significantly greater than that of significantly increased genes, while the number of genes involved in cellular processes,

environmental information processing, human diseases, and organismal systems that significantly increased under starvation was significantly greater than that of significantly decreased genes (Figure 5).

(A)

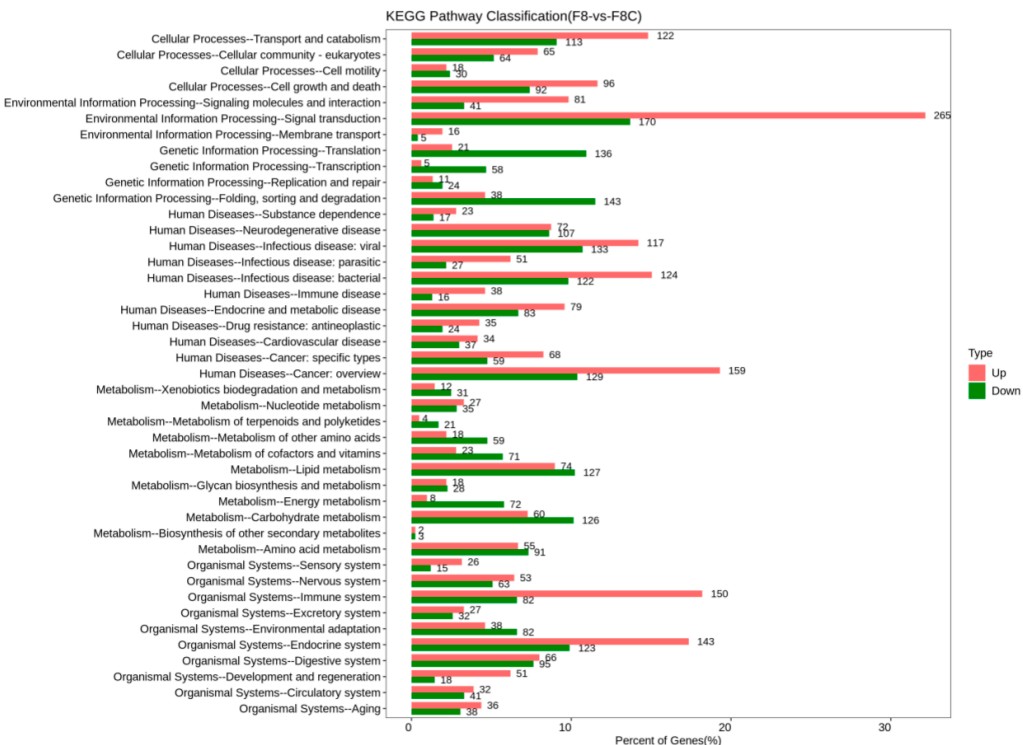

(B)

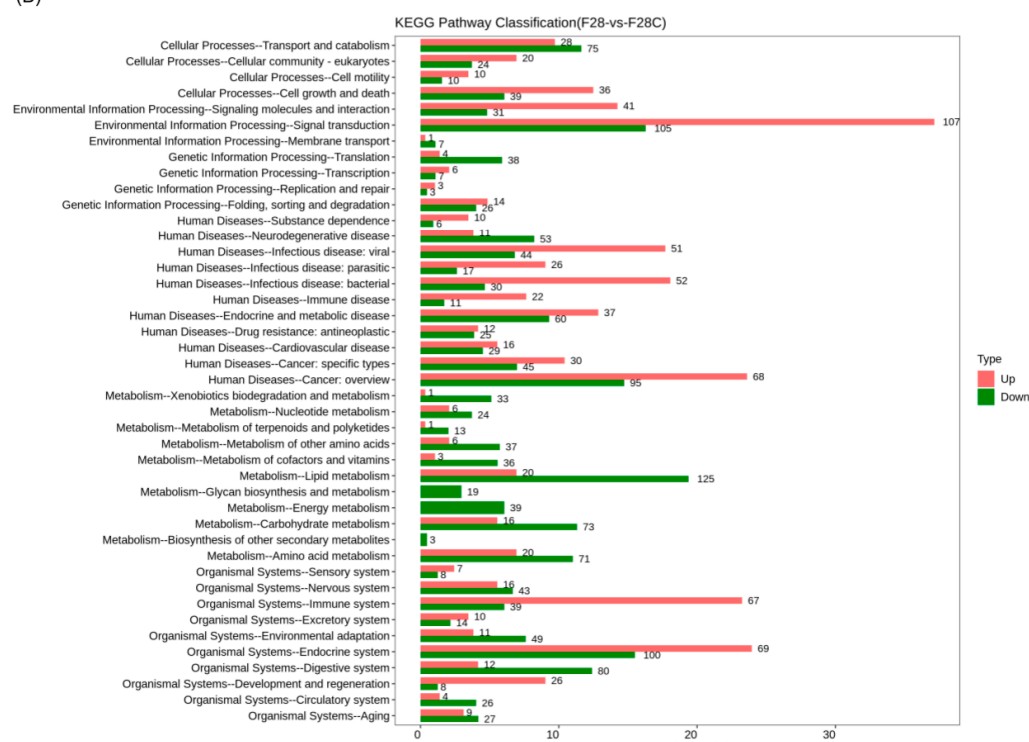

**Figure 5.** Kyoto Encyclopedia of Genes and Genomes pathway classification of significantly upregulated and downregulated genes in the liver transcriptome of yellowcheek starved for (**A**) 8 and (**B**) 28 d compared with the control groups at the same time.

Furthermore, KEGG pathway analysis showed that the transcriptional levels of most genes involved in glycolysis/gluconeogenesis were significantly reduced in yellowcheek starved for 8 d, except for those of aldose 1-epimerase (5.1.3.3), fructose-1,6-bisphosphatase (3.1.3.11), multiple inositol polyphosphate phosphatase 1-like isoform X1 (3.1.3.80), and dihydrolipoamide dehydrogenase (1.8.1.4), which were significantly increased (Figure 6A). However, this trend recovered after 28 d of starvation (Figure 6B). The transcriptional levels of the genes involved in the citrate cycle (Figure 7), fatty acid biosynthesis (Figure 8), pentose phosphate pathway (Figure S2), and N-glycan biosynthesis (Figure S3) exhibited similar trends. Starvation for 8 d did not significantly affect the transcription levels of genes involved in fatty acid elongation in the mitochondria (4 < n < 16). Meanwhile, the transcription levels of most genes involved in fatty acid elongation in the mitochondria (4 < n < 16) were significantly lower in yellowcheek starved for 28 d compared to those in the control group ($p < 0.05$; Figures S4 and S5). However, the transcriptional levels of most of the genes involved in fatty acid elongation in the endoplasmic reticulum (n ≥ 16) were significantly lower in yellowcheek starved for 8 d compared to those of the control, and this reduction was maintained after 28 d of starvation (Figure S6). Compared to the control group, the transcriptional levels of most genes involved in steroid biosynthesis were also significantly lower in yellowcheek starved for 8 d, and the reduction was maintained after 28 d of starvation (Figure S7). It is worth noting that the transcription levels of most upstream genes involved in fatty acid degradation were significantly higher in yellowcheek starved for 8 d than those in the control group, whereas those in yellowcheek starved for 28 d were significantly lower than those in the control group ($p < 0.05$; Figure 9).

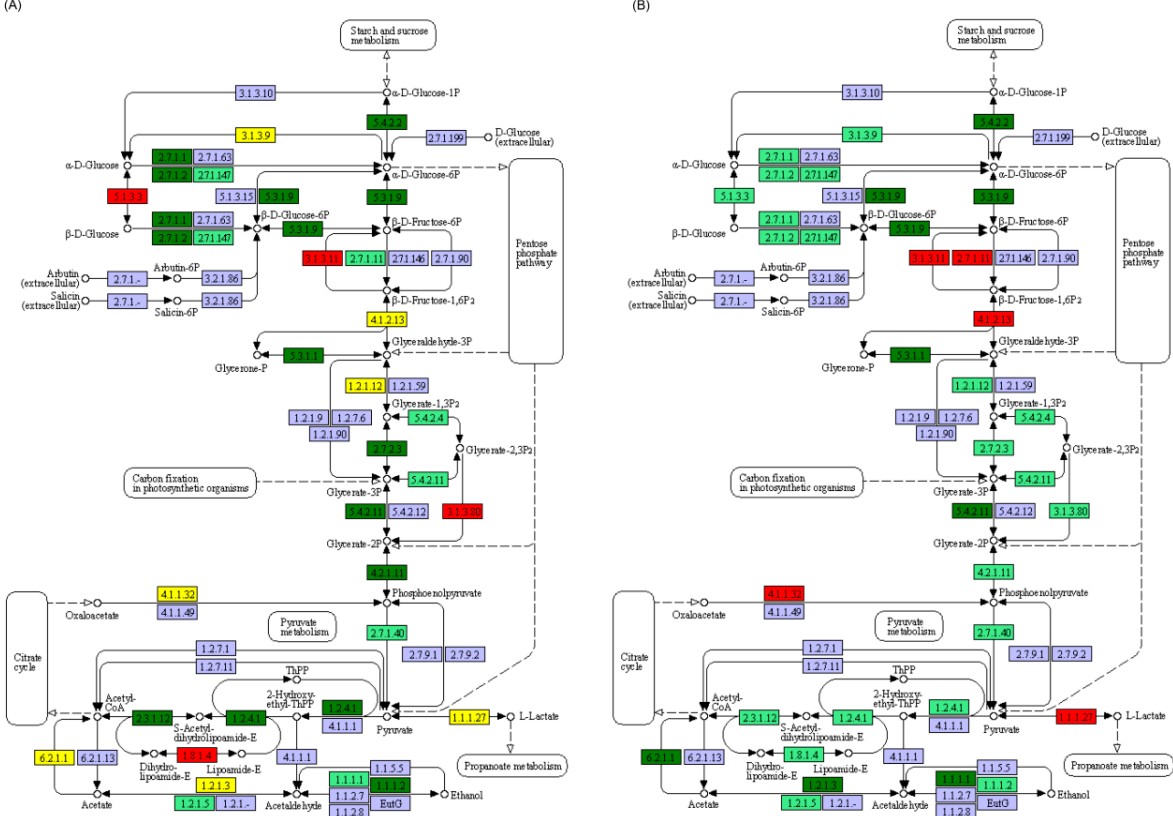

**Figure 6.** Kyoto Encyclopedia of Genes and Genomes pathway showing the significantly upregulated and downregulated genes involved in glycolysis/gluconeogenesis in the liver transcriptome of yellowcheek starved for (**A**) 8 and (**B**) 28 d compared to the control groups at the same time. Red, dark green, yellow, light green, and light purple indicate significantly upregulated genes, downregulated genes, and both upregulated and downregulated genes in the corresponding genes, the genes of the species annotated on the map, and the genes that were not annotated on the map, respectively.

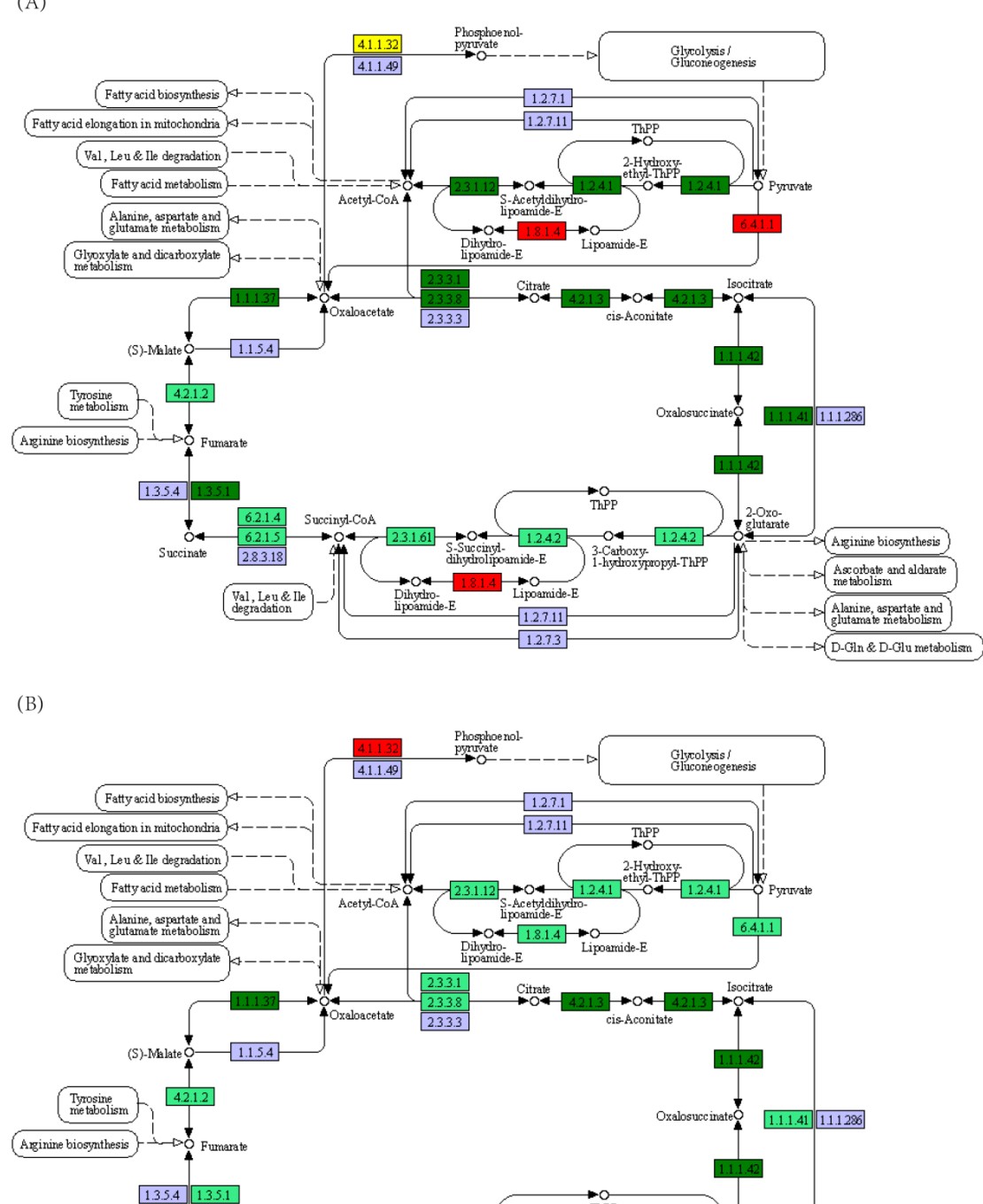

**Figure 7.** Kyoto Encyclopedia of Genes and Genomes pathway showing the significantly upregulated and downregulated genes involved in citrate cycle (TCA cycle) in the liver transcriptome of yellowcheek starved for (**A**) 8 and (**B**) 28 d compared to the control groups at the same time. Red, dark green, yellow, light green, and light purple indicate significantly upregulated genes, downregulated genes, and both upregulated and downregulated genes in the corresponding genes, the genes of the species annotated on the map, and the genes that were not annotated on the map, respectively.

(A)

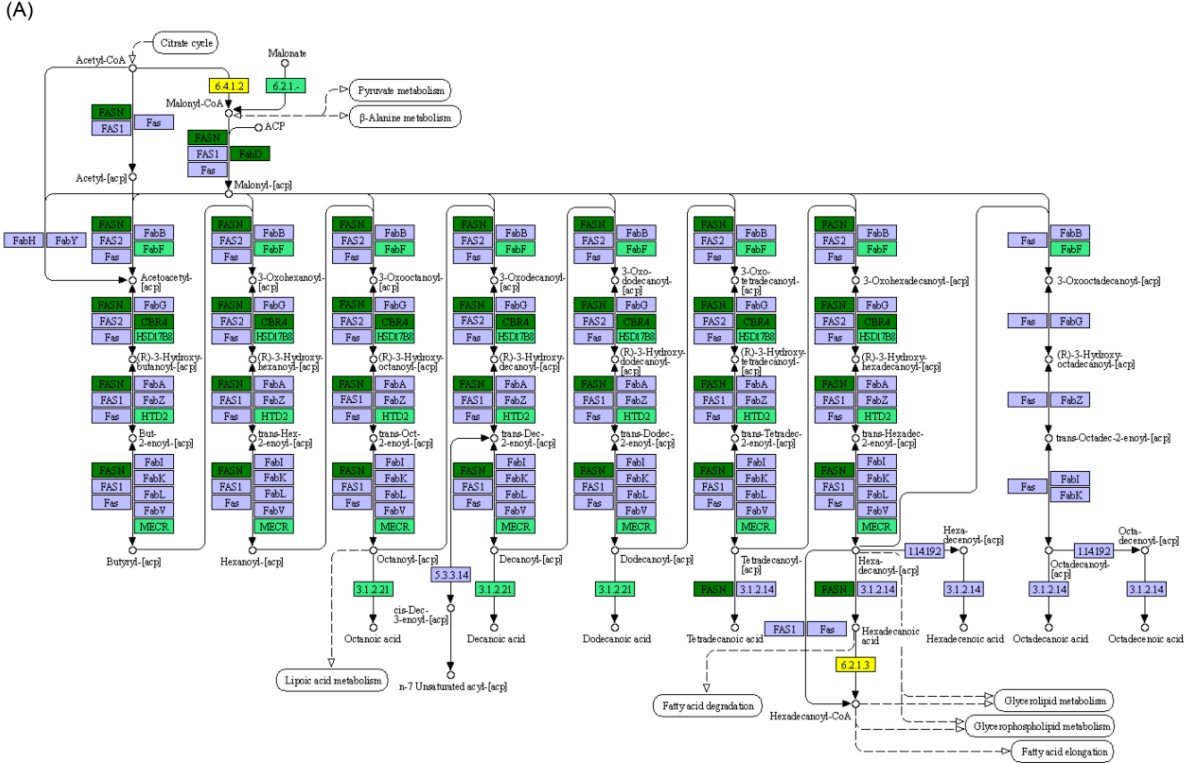

(B)

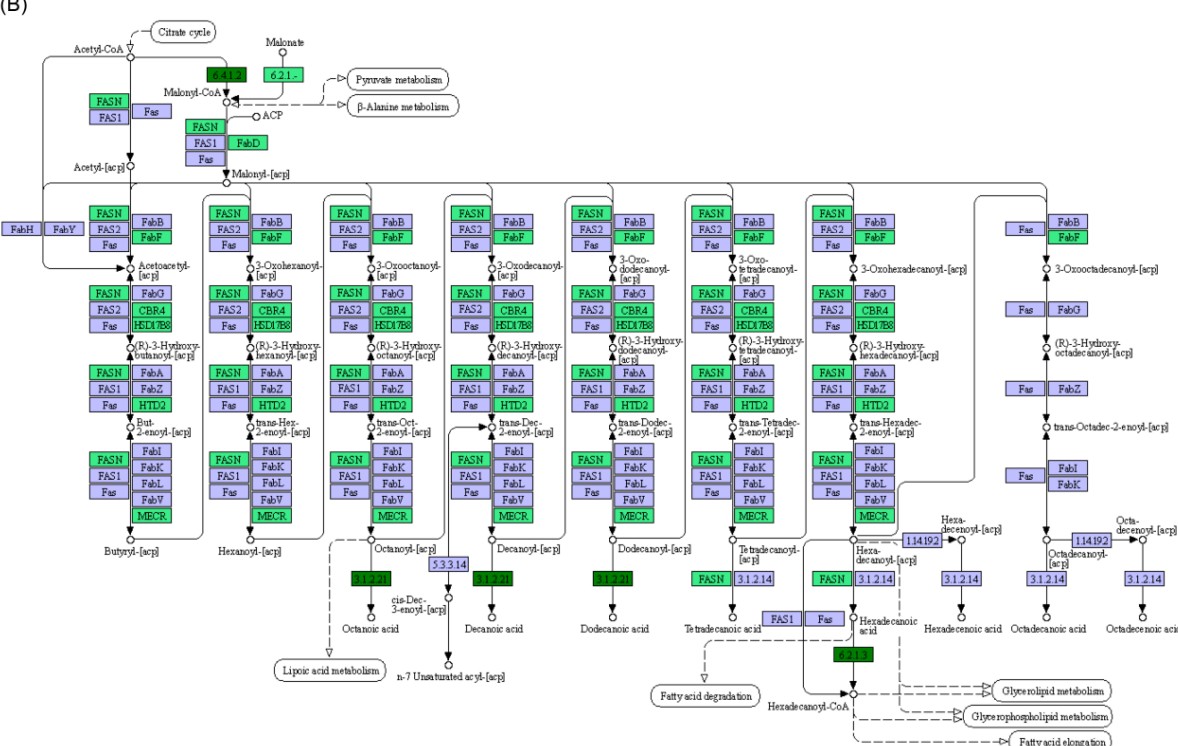

**Figure 8.** Kyoto Encyclopedia of Genes and Genomes pathway showing the significantly upregulated and downregulated genes involved in fatty acid biosynthesis in the liver transcriptome of yellowcheek starved for (**A**) 8 and (**B**) 28 d compared to the control groups at the same time. Dark green, yellow, light green, and light purple indicate significantly downregulated genes, both upregulated and downregulated genes in the corresponding genes, the genes of the species annotated on the map, and the genes that were not annotated on the map, respectively.

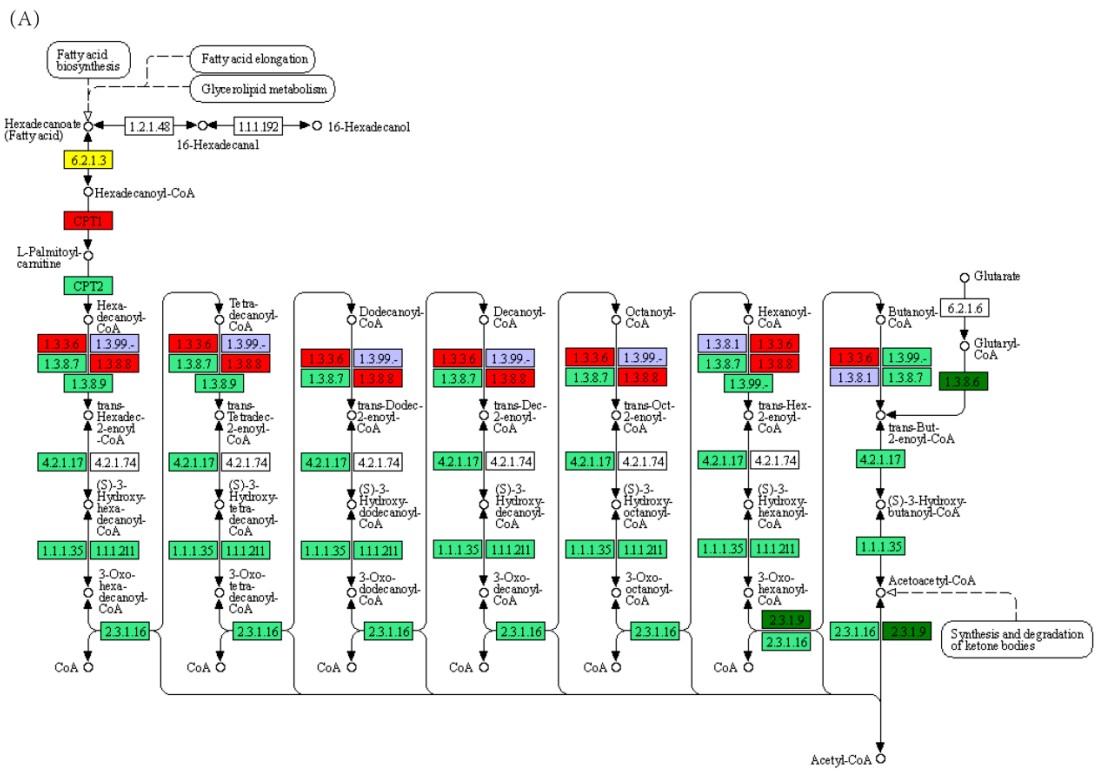

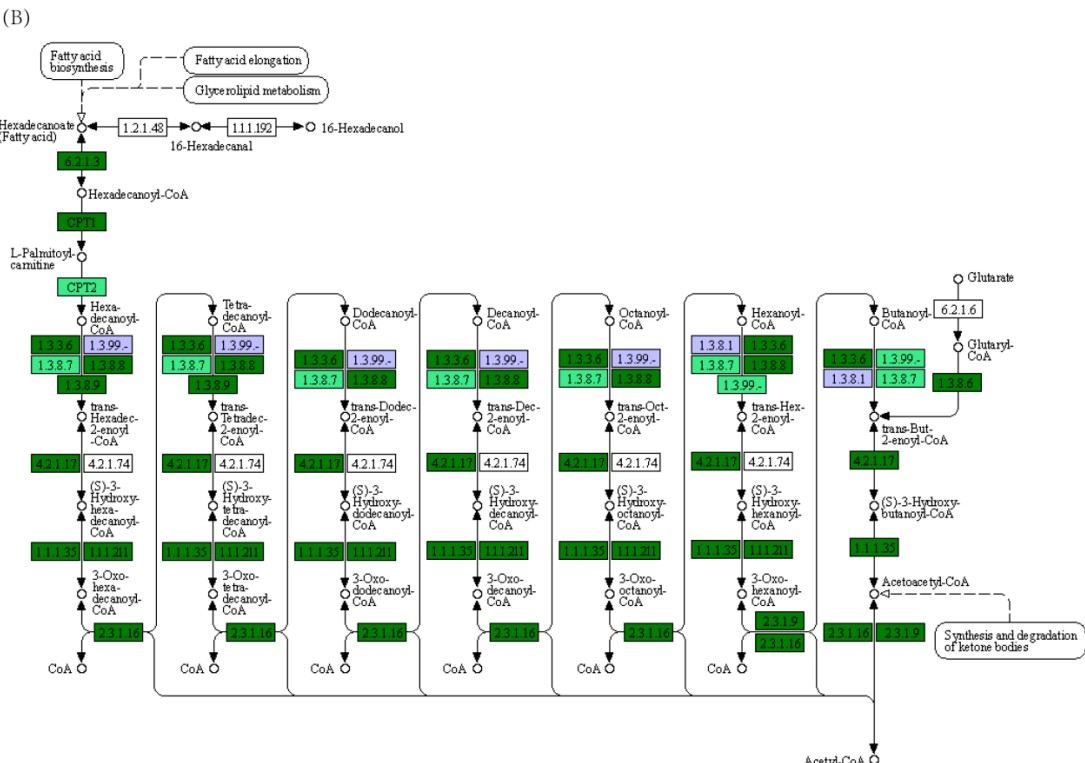

**Figure 9.** Kyoto Encyclopedia of Genes and Genomes pathway showing the significantly upregulated and downregulated genes involved in fatty acid degradation in the liver transcriptome of yellowcheek starved for (**A**) 8 and (**B**) 28 d compared to the control groups at the same time. Red, dark green, yellow, light green, light purple, and white indicate significantly upregulated genes, downregulated genes, and both upregulated and downregulated genes in the corresponding genes, the genes of the species annotated on the map, the genes that were not annotated on the map, and unknown genes, respectively.

## 4. Discussion

Seasonal changes, climate change, and human activities lead to fluctuations in the abundance of natural water or the uneven distribution at the spatiotemporal level, which render the risk of food scarcity and starvation in fishes [27–29]. Starvation has an evident effect on fish growth and metabolism [30–32]. Starvation for 31 d significantly reduced total body weight and standard length of European sea bass (*Dicentrarchus labrax*) compared to controls at the same time, whereas the starvation did not reduce both parameters of blackspot sea bream (*Pagellus bogaraveo*) [33]. Starvation for 30 d also significantly reduced weight gain percent, specific growth ratio, condition factor, and hepatic somatic index of grey mullet (*Mugil cephalus*) with $9.52 \pm 0.98$ cm of total length, whereas it only significantly reduced the condition factor and hepatic somatic index of grey mullet with $12.05 \pm 0.86$ cm of total length [34]. Ölmez et al. [35] conducted a 70 day starvation period to determine the effects of long-term starvation on the whole-body polyunsaturated fatty acid composition and fatty acid metabolism-related gene expression in the liver of zebrafish (*Danio rerio*). Their results showed that the expression of seven (*elvol5*, *fads2*, *cpt1-β*, *acox1*, *acadvl*, *fabp1a*, and *fabp7a*) genes was significantly downregulated towards the end of the starvation period, suggesting that the mRNA expression of genes involved in fatty acid metabolism was negatively influenced by long-term starvation. Li et al. [29] reported that liver glycogen levels decreased in Gibel carp (*Carassis auratus gibelio*) starved for 7 d, whereas muscle glycogen levels decreased in the group starved for 21 d, suggesting that glycogen in the liver is first utilized to provide energy rather than that in the muscle. Tripathi and Verma [30] reported that starvation gradually decreased the activity of citrate synthase, glucose-6-phosphate dehydrogenase, and lactate dehydrogenase in the brain, liver, and skeletal muscle of the freshwater catfish *Clarias batrachus*, indicating a substantial decline in aerobic and biosynthetic capacities during starvation. Our results showed that starvation significantly reduced the body weight, K, visceral weight, and viscera index of yellowcheek, and the significant reduction in visceral weight and viscera index occurred earlier than the reduction in body weight and K, suggesting that yellowcheek prefers glycogen as an energy source, rather than muscle protein, under starvation.

Starvation induces different responses on blood parameters depending on how long the starvation lasts and on the species-specific differences in the metabolism [36]. Starvation for 7 d significantly decreased plasma GLU levels, whereas it significantly increased plasma free fatty acids levels of Gibel carp [26]. Starvation for 14 d significantly reduced blood hemoglobin, hematocrite, protein, creatinine, TG, and electrolyte sodium of *Notopterus notopterus*, whereas it significantly increased blood GLU and urea nitrogen [36]. Starvation for 30 d significantly reduced serum GLU, cholesterol, cortisol, and STP of grey mullet with $9.52 \pm 0.98$ cm of total length, whereas it did not reduce these parameters of grey mullet with $12.05 \pm 0.86$ cm of total length [34]. Moreover, plasma GLU, cholesterol, and peroxidase contents were significantly lower in starved tinfoil barb (*Barbonymus schwanenfeldii*) for 2 weeks compared to those in fed fish [37]. Our results showed that starvation for 28 d significantly reduced the serum GLU, STP, globulin, ALB, and HDL. Moreover, from starvation for 8 d, the serum ALT, AST, TG, TC, and ALP levels were significantly reduced compared to the control groups at the same time. These different responses are most likely caused by the species-specific differences in the metabolism.

Transcription and metabolic characteristics of the liver, an important metabolic organ, are affected by fish growth, health status, and habitat conditions [38–40]. Li et al. [29] reported that the transcriptional levels of glucose transporter type 2 (GLUT2) in the liver of Gibel carp were upregulated during starvation, whereas no changes in the mRNA levels of glycolytic enzymes (glucokinase and 6-phosphofructokinase) were observed. Moreover, they found that the mRNA levels of gluconeogenic enzymes, including glucose-6-phosphatase (G6Pase), fructose 1,6-bisphosphatase (FBPase), and phosphoenolpyruvate carboxykinase (PEPCK), were enhanced during starvation, indicating that gluconeogenic potential increased with starvation, possibly for blood glucose homeostasis [29]. Furthermore, they found that mRNA levels of carnitine palmitoyl transferase 1 isoform a

(CPT1a) and acyl-CoA oxidase 3 (ACO3) were upregulated in the liver during starvation, suggesting that fatty acids are catabolized to provide energy during starvation [29]. In our study, starvation significantly changed the liver transcriptome of yellowcheek, and different starvation levels (8 d vs. 28 d) had different effects on the liver transcriptome (Figure 3B). This difference may be caused by the different nutrients used to produce energy by yellowcheek at different stages of starvation. Moreover, the transcriptional levels of most genes involved in glycolysis/gluconeogenesis, citrate cycle, fatty acid biosynthesis, pentose phosphate, and N-glycan biosynthesis were significantly reduced in yellowcheek starved for 8 d, whereas this trend was recovered after 28 d of starvation. Although starvation for 8 d did not significantly affect the transcription levels of genes involved in fatty acid elongation in the mitochondria ($4 < n < 16$) in yellowcheek livers, the transcription levels of most genes involved in fatty acid elongation in the mitochondria ($4 < n < 16$) in yellowcheek livers, after 28 d of starvation, were significantly lower than those in the control. However, the transcriptional levels of most of the genes involved in fatty acid elongation in the endoplasmic reticulum ($n \geq 16$), in the liver of yellowcheek starved for 8 d, were significantly lower than those of the control, and this reduction was maintained after 28 d of starvation. Moreover, the transcription levels of most upstream genes involved in fatty acid degradation in the liver of yellowcheek after 8 d of starvation were significantly higher than those in the control, whereas the transcription levels of most genes involved in fatty acid degradation were significantly lower than those in the control after 28 d of starvation. These results suggest that short-term starvation (8 d) limited the biosynthesis of N-glycan and fatty acids, as well as fatty acid elongation in the endoplasmic reticulum ($n \geq 16$) and upregulated fatty acid degradation in the liver of yellowcheek, which was coincident with the results of Machado et al. [4] on catfish. However, long-term starvation (28 d) alleviated the reduction in N-glycan and fatty acid biosynthesis caused by early starvation, and it significantly reduced fatty acid elongation in the mitochondria ($4 < n < 16$), as well as fatty acid degradation. This may be an underlying mechanism by which all freshwater carnivorous fish cope with starvation.

Generally, re-feeding after starvation is used to evaluate whether the effect of starvation on fish is reversible [1,3]. However, due to the limitations in the experimental conditions, this study did not conduct a re-feeding experiment after starvation. An in-depth study of the effect of re-feeding, after starvation, on the growth and metabolism of yellowcheek will provide significant information for the protection of the natural yellowcheek population and artificial culture of the fish. Nevertheless, our results still provide important reference information for assessing the starvation levels of wild yellowcheek.

## 5. Conclusions

Short-term starvation (8 d) did not cause significant changes in body weight, body length, and K value of yellowcheek, whereas it significantly reduced the visceral weight and viscera index. Meanwhile, long-term starvation (28 d) caused a significant reduction in the body weight and K value of yellowcheek, and it maintained the significant reduction in visceral weight and viscera index. These results suggest that yellowcheek utilize glycogen as an energy source, rather than muscle protein, under starvation. Short-term starvation limited the biosynthesis of N-glycan and fatty acids, fatty acid elongation in the endoplasmic reticulum ($n \geq 16$), and upregulated fatty acid degradation. However, long-term starvation alleviated the reduction in N-glycan and fatty acid biosynthesis in yellowcheek liver, caused by early starvation, and significantly reduced fatty acid elongation in the mitochondria ($4 < n < 16$), as well as fatty acid degradation. This may be a mechanism by which all freshwater carnivorous fish cope with starvation.

**Supplementary Materials:** The following supporting information can be downloaded at: https://www.mdpi.com/article/10.3390/fishes8040175/s1, Figure S1: Volcano plots show differentially expressed genes caused by starvation for 8 (A) and 28 (B) days; Figure S2: KEGG pathway showed that significantly upregulated and downregulated genes participate in pentose phosphate in the liver transcriptome of yellowcheek starved for 8 (A) and 28 (B) days compared with the control;

Figure S3: KEGG pathway showed that significantly upregulated and downregulated genes participate in N-glycan biosynthesis in the liver transcriptome of yellowcheek starved for 8 (A) and 28 (B) days compared with the control; Figure S4: KEGG pathway showed that significantly upregulated and downregulated genes participate in fatty acid elongation in mitochondria (4 < n < 16) in the liver transcriptome of yellowcheek starved for 8 (A) and 28 (B) days compared with the control; Figure S5: KEGG pathway showed that significantly upregulated and downregulated genes participate in general forms of fatty acid elongation in mitochondria (4 < n < 16) in the liver transcriptome of yellowcheek starved for 8 (A) and 28 (B) days compared with the control; Figure S6: KEGG pathway showed that significantly upregulated and downregulated genes participate in fatty acid elongation in endoplasmic reticulum (n ≥ 16) in the liver transcriptome of yellowcheek starved for 8 (A) and 28 (B) days compared with the control; Figure S7: KEGG pathway showed significantly upregulated and downregulated genes participate in steroid biosynthesis in the liver transcriptome of yellowcheek starved for 8 (A) and 28 (B) days compared with the control; Table S1: Basic information of sequencing data.

**Author Contributions:** Conceptualization, M.X. and G.Z.; methodology, M.X., S.L., Z.F. and Q.D.; software, M.X. and P.W.; validation, J.X., H.W. and J.G.; formal analysis, M.X., S.L. and J.X.; investigation, M.X., S.L., Z.F., P.W., H.W., J.G., G.Z. and G.X.; resources, M.X., J.X. and G.Z.; data curation, P.W. and G.X.; writing—original draft preparation, M.X.; writing—review and editing, J.X. and G.Z.; visualization, M.X. and P.W.; supervision, J.X. and G.Z.; project administration, G.Z.; funding acquisition, M.X. and G.Z. All authors have read and agreed to the published version of the manuscript.

**Funding:** This research was funded by the Changsha Natural Science Foundation for Changsha Science and Technology Bureau, grant number kq2202355, and the Earmarked Fund for China Agriculture Research System (CARS-45).

**Institutional Review Board Statement:** The animal research was approved by the Animal Care Committee of the Hunan Fisheries Science Institute (No. HFSI2021-03). All experimental phases were strictly controlled.

**Informed Consent Statement:** Not applicable.

**Data Availability Statement:** The raw transcriptome data were submitted to the NCBI Gene Expression Omnibus (GEO) database with accession number 23700447.

**Acknowledgments:** We would like to thank Jiajia Ni at Guangdong Meilikang Bio-Science Ltd., China for assistance with data analysis and manuscript revision.

**Conflicts of Interest:** The authors declare no conflict of interest.

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
