# Peer review of "Effects of Starvation on the Physiology and Liver Transcriptome of Yellowcheek (Elopichthys bambusa)"

_fishes, doi:10.3390/fishes8040175_

Round 1
Reviewer 1 Report
Dear Authors,
The authors effects of starvation (short (8) and long term (28d)) on the physiology and liver transcriptome of yellowcheek. They investigated biochemical and molecular analysis in liver tissues of the fish. Results of the paper are potentially interesting for “fishes” and it is well aimed. Before decision on the manuscript, the author should give appropriate answer to comments at the below.
Minor comments
The authors should add details about some kit (company, city and country).
The authors did not provide information about the sex of the fish. this issue may be important for the visceral index and some serum parameters. Especially in some stages of oogenesis (vitellogenesis), important changes are observed in serum parameters. Vitellogenin, secreted from the liver and transported through the blood, is also transported to the oocytes by endocytosis. Therefore, I think the sex of the fish is important for this study.
Author Response
Responses to the reviewer’s comments
Minor comment
The authors should add details about some kit (company, city and country).
Responses
Thank you for your comment. We have added details about some kits (company, city and country) according to your comment.
Minor comments
The authors did not provide information about the sex of the fish. this issue may be important for the visceral index and some serum parameters. Especially in some stages of oogenesis (vitellogenesis), important changes are observed in serum parameters. Vitellogenin, secreted from the liver and transported through the blood, is also transported to the oocytes by endocytosis. Therefore, I think the sex of the fish is important for this study.
Response
Thank you for your comments. We fully agree with your comments on the impact of sex on the visual index, serum parameters, and liver transcription and metabolism of fish. However, the fish we studied were too small to distinguish sex. Their gonads were not developed, and there is no morphological difference between different sexes of the fish. Therefore, we cannot distinguish the sex of the fish we studied, and only randomly collected samples for analysis.
Reviewer 2 Report
Review of Xie et al. Elopichthys
This study deals with an experiment on the effects of starvation on the yellow cheek and in particular the transcriptome of the liver. Several studies on the effects of starvation on metabolism in fishes have been carried out before, but the present study exceeds them as the entire transcriptomes of liver from starved and fed fish were studied.
The motivation is very weak. I cannot see how the knowledge produced could be used in preserving the yellow cheek (that starvation is bad is not surprising). If the aim is to study fish physiology in general a species with more background information would have been a better choice.
Experiment carried out outdoors? Mortality during the experiment? What and how much were the fish fed before the experiment?
Most important: What was the level of feeding for the controls? Satiation? Ascertaining that fish are really fed to satiation would demand that the fish are fed so much that a little food is uneaten. Some percentage of body weight? The lack of information on the feeding levels in controls is a major limitation of the study.
Line 86. Five fish from each group sampled. 4 treatments (control-starvationX8 and 28 days) and 3 replicates. = 4x3x5=60, 360 fish were used. What about the rest?
Viscera weight includes which organs? Why was not the liver weighed separately?
Line 130. Results with p < 0.05 were considered significant, which is appropriate. However, I think that it could be useful for the main results to show at least what reaches p 0.01 and 0.001 so that one can see what is highly and not just barely significant.
Figure 1. It should be possible to understand a figure from the information in the legends without searching the general text. Explain F8C, F28C, F8 and F28.
Figure 2. There are very large differences between controls at different points of time in A, B and C. I suppose that one would aim at keeping the controls under a regime where they did not change much over the period, but rather kept the original levels, but this is not at all the case here. What has happened?
Line 200-201. The changes after 8 and 28 days of starvation were similar, which is also supported by Figure 3B. However, the transcription profile in controls at days 8 and 28 are clearly separated (3B). This should at least be discussed. It should also be pointed out in the legends for Figs 4 and 6 that the results are compared with the controls at the same time (which I suppose they are and which they should be).
Author Response
Comment
This study deals with an experiment on the effects of starvation on the yellow cheek and in particular the transcriptome of the liver. Several studies on the effects of starvation on metabolism in fishes have been carried out before, but the present study exceeds them as the entire transcriptomes of liver from starved and fed fish were studied.
Response
Thank you very much for reviewing our manuscript and providing the very useful comments. The comments are very important for us to revise our manuscript. We have revised our manuscript according to your and the other reviewers’ comments.
Comment
The motivation is very weak. I cannot see how the knowledge produced could be used in preserving the yellow cheek (that starvation is bad is not surprising). If the aim is to study fish physiology in general a species with more background information would have been a better choice.
Response
Thank you for your comment. The purposes of this study are to clarify the effect of starvation on the metabolism of yellowcheek, and to find the serial parameters and liver transcriptional markers used to evaluate the different starvation levels of yellowcheek. We have added the description in the Introduction of our revised manuscript.
Comment
Experiment carried out outdoors? Mortality during the experiment? What and how much were the fish fed before the experiment?
Response
The experiment was carried out outdoors. No fish died during the experiment. The dry weight of the daily feed was 3% of the average body weight of the fish before the experiment and during the acclimatize stage. We have added the information in our revised manuscript.
Comments
Most important: What was the level of feeding for the controls? Satiation? Ascertaining that fish are really fed to satiation would demand that the fish are fed so much that a little food is uneaten. Some percentage of body weight? The lack of information on the feeding levels in controls is a major limitation of the study.
Response
Thank you very much for your comment. This is a very important issue for our experiment. The experimental fish were randomly and equally assigned to 12 outdoor cement pools to acclimatize for 1 week before formal experiment and fed commercial feed twice daily (8:00 and 18:00). The dry weight of the daily feed was 3% of the average body weight of the fish as before experiment. During the formal experiment, the control groups were continuously fed a commercial diet as acclimatize stage. We have added the information in our revised manuscript. Thank you again.
Comments
Line 86. Five fish from each group sampled. 4 treatments (control-starvationX8 and 28 days) and 3 replicates. = 4x3x5=60, 360 fish were used. What about the rest?
Responses
Yellowcheek needs 4-5 years to reach sexual maturity, while our experimental fish was 1 year old. The rest fish were re-fed to study the effects of starvation at the developmental stage on the gonadal development and fecundity of yellowcheek after adulthood. Because the development time to mature is too long, we have not obtained the results of this experiment.
Comments
Viscera weight includes which organs? Why was not the liver weighed separately?
Responses
Viscera weight mainly includes liver and intestinal weight. We have added the results of the liver weighed separately in our revised manuscript.
Comments
Line 130. Results with p < 0.05 were considered significant, which is appropriate. However, I think that it could be useful for the main results to show at least what reaches p 0.01 and 0.001 so that one can see what is highly and not just barely significant.
Responses
Thank you for your comment. Although results with p < 0.05 were considered significant, we also distinguished the case of p < 0.05, p < 0.01 and p < 0.001 using *, **, and ** in the figures.
Comment
Figure 1. It should be possible to understand a figure from the information in the legends without searching the general text. Explain F8C, F28C, F8 and F28.
Response
Thank you for your comment. We have explain F8C, F28C, F8 and F28 in our revised manuscript according to your comment.
Comment
Figure 2. There are very large differences between controls at different points of time in A, B and C. I suppose that one would aim at keeping the controls under a regime where they did not change much over the period, but rather kept the original levels, but this is not at all the case here. What has happened?
Response
Thank you for your comment. We don’t know exactly what happened. This may be due to the differences in these parameters caused by some unobserved changes in the water environment as the experiment conducted outdoor, or due to the differences in sample collection and preservation. However, we carried out the measurement in strict accordance with the test standards of various parameters.
Comment
Line 200-201. The changes after 8 and 28 days of starvation were similar, which is also supported by Figure 3B. However, the transcription profile in controls at days 8 and 28 are clearly separated (3B). This should at least be discussed. It should also be pointed out in the legends for Figs 4 and 6 that the results are compared with the controls at the same time (which I suppose they are and which they should be).
Response
Thank you very much for your comments. We have added the discussion about the difference. Moreover, we also have pointed out in the legends for Figs -9 that the results are compared with the controls at the same time according to your comments.
Reviewer 3 Report
The work presented by Xie, et al. on “Effects of starvation on the physiology and liver transcriptome of yellowcheek (Elopichthys bambusa)” is interesting and enhance the present understanding of starvation effect on carnivorous fish. However, I could see several flaws that need to be addressed. Overall, the English language is poor and needs to be improved.
My comments for improvement of the MS are noted below:
Abstract:
In the abstract, the application of the findings in aquaculture and the further research areas in the line may be included.
Introduction
The information in the introduction part is very short. Many studies have been carried out in the past. The relevant finding in the line of effect of starvation on carnivorous fish has to be included.
Materials and Methods
The fish rearing conditions including water quality and other management aspects may be elaborated.
In the control group, what was the quantity of feed given needs to be mentioned
Results and Discussion
The analysis of obtained data needs to be reflected through improved way of representation.
Nothing has been discussed about the alteration of serum parameters which needs to be discussed with relevant justification.
Elaborate discussion on the effect of starvation on growth is necessary.
Very little research work has been cited in the discussion part of the work. More research paper needs to be studied in the line and correlated with the findings.
Line No 323-325: reference may be cited
The discussion part is short, which may be broadened.
Discussion should be able to extend comprehensive knowledge for further work which needs rewriting.
Author Response
Comments and suggestions for authors
The work presented by Xie, et al. on “Effects of starvation on the physiology and liver transcriptome of yellowcheek (Elopichthys bambusa)” is interesting and enhance the present understanding of starvation effect on carnivorous fish. However, I could see several flaws that need to be addressed. Overall, the English language is poor and needs to be improved.
Responses
Thank you very much for reviewing our manuscript and providing the very useful comments. The comments are very important for us to revise our manuscript. We have revised our manuscript according to your and the other reviewers’ comments. Moreover, the language of our manuscript had been revised through a native English expert in the Editage (https://www.editage.cn/).
Comment
Abstract:
In the abstract, the application of the findings in aquaculture and the further research areas in the line may be included.
Response
Thank you for your comment. We have revised the Abstract according to your comment. However, due to the limitation of the number of words, we cannot elaborate in detail and comprehensively according to your comments.
Comment
Introduction
The information in the introduction part is very short. Many studies have been carried out in the past. The relevant finding in the line of effect of starvation on carnivorous fish has to be included.
Responses
Thank you for your comment. We have added more introduction of starvation on fish physiology and metabolism according to your comment.
Comments
Materials and Methods
The fish rearing conditions including water quality and other management aspects may be elaborated.
In the control group, what was the quantity of feed given needs to be mentioned
Responses
The fish were randomly and equally assigned to 12 outdoor cement pools with 0.8 m of water depth to acclimatize for 1 week before formal experiment and fed commercial feed twice daily (8:00 and 18:00). The dry weight of the daily feed was 3% of the average body weight of the fish as before the experiment. The control groups were continuously fed a commercial diet (F8C and F28C) as acclimatize stage. During the experiment, the water temperature, pH, and dissolved oxygen were 23.5±3.4 (from 20.2 to 27.6)°C, 6.9±0.2 (from 6.7 to 7.2), and 6.1±0.3 (from 5.7 to 6.5) mg/L, respectively. A small aerator was used in each cement pool to continuously increase oxygen during the experiment. We have added the description of breeding conditions according to your comments.
Comments
Results and Discussion
The analysis of obtained data needs to be reflected through improved way of representation.
Nothing has been discussed about the alteration of serum parameters which needs to be discussed with relevant justification.
Elaborate discussion on the effect of starvation on growth is necessary.
Very little research work has been cited in the discussion part of the work. More research paper needs to be studied in the line and correlated with the findings.
Line No 323-325: reference may be cited
The discussion part is short, which may be broadened.
Discussion should be able to extend comprehensive knowledge for further work which needs rewriting.
Responses
Thank you very much for your comments. We have expanded the discussion section of our manuscript and added the discussion on the impact of starvation on serum parameters according to your comments.
Round 2
Reviewer 3 Report
May be accepted in the revised form